# Differentiation of Small Clear Renal Cell Carcinoma and Oncocytoma through Magnetic Resonance Imaging-Based Radiomics Analysis: Toward the End of Percutaneous Biopsy

**DOI:** 10.3390/jpm13101444

**Published:** 2023-09-28

**Authors:** Thibault Toffoli, Olivier Saut, Christele Etchegaray, Eva Jambon, Yann Le Bras, Nicolas Grenier, Clément Marcelin

**Affiliations:** 1Centre Hospitalier Universitaire (CHU) de Bordeaux, Imaging and Interventional Radiology, Hôpital Pellegrin, 33000 Bordeaux, France; t.toffoli47@gmail.com (T.T.); eva.jambon@chu-bordeaux.fr (E.J.); yann.lebras@chu-bordeaux.fr (Y.L.B.); 2University of Bordeaux, IMB, UMR CNRS 5251, INRIA Project Team Monc, F-33400 Talence, France; olivier.saut@inria.fr (O.S.); christele.etchegaray@inria.fr (C.E.); nicolas.grenier@chu-bordeaux.fr (N.G.); 3Bordeaux Institute of Oncology, BRIC U1312, INSERM, Bordeaux University, 33000 Bordeaux, France

**Keywords:** MRI, renal tumors, radiomics

## Abstract

Purpose: The aim of this study was to ascertain whether radiomics data can assist in differentiating small (<4 cm) clear cell renal cell carcinomas (ccRCCs) from small oncocytomas using T2-weighted magnetic resonance imaging (MRI). Material and Methods: This retrospective study incorporated 48 tumors, 28 of which were ccRCCs and 20 were oncocytomas. All tumors were less than 4 cm in size and had undergone pre-biopsy or pre-surgery MRI. Following image pre-processing, 102 radiomics features were evaluated. A univariate analysis was performed using the Wilcoxon rank-sum test with Bonferroni correction. We compared multiple radiomics pipelines of normalization, feature selection, and machine learning (ML) algorithms, including random forest (RF), logistic regression (LR), AdaBoost, K-nearest neighbor, and support vector machine, using a supervised ML approach. Results: No statistically significant features were identified via the univariate analysis with Bonferroni correction. The most effective algorithm was identified using a pipeline incorporating standard normalization, RF-based feature selection, and LR, which achieved an area under the curve (AUC) of 83%, accuracy of 73%, sensitivity of 79%, and specificity of 65%. Subsequently, the most significant features were identified from this algorithm, and two groups of uncorrelated features were established based on Pearson correlation scores. Using these features, an algorithm was established after a pipeline of standard normalization and LR, achieving an AUC of 90%, an accuracy of 77%, sensitivity of 83%, and specificity of 69% for distinguishing ccRCCs from oncocytomas. Conclusions: Radiomics analysis based on T2-weighted MRI can aid in distinguishing small ccRCCs from small oncocytomas. However, it is not superior to standard multiparameter renal MRI and does not yet allow us to dispense with percutaneous biopsy.

## 1. Introduction

Renal cell carcinomas (RCCs) rank among the top ten most prevalent tumors, accounting for 140,000 deaths annually worldwide, with a growing incidence [1]. Renal tumors encompass a broad spectrum of neoplasms, exhibiting variations in clinical behavior, histopathologic characteristics, and genetic expressions.

A large number of RCC cases are incidentally diagnosed, as the disease tends to remain asymptomatic for a considerable duration. The increased use of cross-sectional imaging has led to a surge in the detection of small renal masses (SRMs), characterized as contrast-enhancing masses with a maximum diameter of 4 cm [2,3]. SRMs constitute nearly 40% of all diagnosed renal tumors [4]. Approximately 20% of these masses are benign, while a substantial proportion of malignant SRMs exhibit slow growth kinetics and non-aggressive histologic features, spurring interest in less invasive treatments compared to surgery [2].

Percutaneous renal biopsy stands as a valuable procedure, offering a pathological diagnosis for renal masses, with an overall diagnostic rate of 92% [5]. This diagnostic insight is crucial for clinicians in making informed decisions about the management and treatment of patients with renal masses. However, it is important to recognize that renal biopsy is not without its complexities and limitations. Firstly, it is worth noting that renal biopsy is an invasive procedure, and there are inherent risks and potential complications. Patients undergoing renal biopsy may experience bleeding, infection, or injury to surrounding structures. While these complications are relatively rare, they underscore the need for careful consideration before performing the procedure. One significant challenge associated with renal biopsy is the risk of sampling errors and non-diagnostic analysis, which can occur in up to 20% of cases [6,7,8]. This means that, despite the invasive nature of the procedure and potential risks to the patient, there is still a significant possibility that the obtained tissue samples may not provide sufficient diagnostic information. Such limitations can result from various factors, including inadequate sample size or sampling of non-representative areas within the renal mass. These issues can lead to uncertainty in diagnosis and may necessitate the repetition of the biopsy or reliance on additional diagnostic methods. Moreover, distinguishing between certain benign tumors, such as oncocytomas, and malignant ones through percutaneous biopsy can be challenging. This challenge is exemplified by the findings of Patel et al., who reported a predictive positive value of only 67% for percutaneous biopsy [9]. This relatively low positive predictive value underscores the difficulty in definitively characterizing some renal masses solely through biopsy. The consequences of misdiagnosis in such cases can be significant, potentially leading to inappropriate treatment decisions. In light of these complexities and limitations associated with percutaneous renal biopsy, there is a growing need for complementary diagnostic approaches, such as radiomics, to enhance the accuracy and reliability of renal mass characterization. Radiomics offers the potential to provide additional quantitative and qualitative information from medical images, aiding in the differentiation of benign and malignant masses and reducing the reliance on invasive procedures with associated risks and diagnostic uncertainties

Conventional non-invasive methods, such as dynamic contrast-enhanced computed tomography (CT) and contrast-enhanced magnetic resonance imaging (MRI), offer qualitative assessment for characterizing renal masses [10]. Specifically, multiparameter MRI has been extensively used in the evaluation of common RCC subtypes (e.g., clear cell RCC (ccRCC), papillary RCC, and chromophobe RCC), combining T2-weighted imaging, diffusion-weighted imaging, chemical shift sequences like integrative personal omics profiling sequences, and post-contrast T1-weighted imaging [10].

Angiomyolipoma and oncocytoma are the two most benign solid renal lesions. While angiomyolipoma can be readily distinguished from other renal tumors [11], oncocytomas, accounting for 5–7% of all primary renal neoplasms [12], exhibit imaging characteristics overlapping with common RCC subtypes, especially ccRCCs. This overlap results in poor imaging diagnostic accuracy [10,13].

In an attempt to overcome the challenges presented by conventional imaging, a new field, known as radiomics, has emerged in recent years. Radiomics aims to extract maximum information from images using machine learning (ML) and deep learning algorithms, surpassing the capability of standard radiological analysis [14].

Radiomics offers several advantages for characterizing small renal masses: early detection, tumor differentiation, and predictive precision. By better understanding tumor biology through radiomics, it becomes possible to personalize treatments, avoiding unnecessary surgery for benign masses.

Numerous studies have employed radiomics in renal applications to distinguish benign from malignant tumors [15,16,17], determine malignant tumor subtypes [18], evaluate biological aggressiveness [19], and detect genetic mutations [20].

The goal of our study was to assess the diagnostic value of T2-weighted MRI-based radiomics features in distinguishing small ccRCCs and oncocytomas.

## 2. Materials and Methods

### 2.1. Study Design

We conducted a single-center, retrospective study using radiology databases and medical records from our institution. We examined 28 cases of histologically verified ccRCCs and 20 histologically verified oncocytomas from 2011 to 2020. All selected cases had a tumor size of less than 4 cm and had undergone pre-surgery or biopsy MRI. Patients without a pre-surgery or biopsy MRI or those with poor exam quality were excluded. Consequently, the study included 20 patients with oncocytomas and 28 patients with ccRCCs (Figure 1).

### 2.2. MRI Protocol

All MRI procedures were performed using an Achieva 1.5T scanner (Philips Medical Systems, Amsterdam, The Netherlands) or 1.5T Avanto scanner (SIEMENS, Erlingen, Germany), and a Discovery 3T MR750W scanner (GE Medical Systems, Chicago, IL, USA). Each patient underwent a standard protocol incorporating T2-weighted, chemical shift in-phase and out-phase, diffusion-weighted imaging, and post-contrast T1-weighted imaging axial sequences. The MRI parameters are presented in Table 1.

### 2.3. Radiomics Processing

Initially, we extracted DICOM data from our medical database and anonymized the patients using Horos. A radiologist (T.T.) manually conducted three-dimensional segmentation of tumors using LIFEx freeware. Following this, we applied an N4 artifact denoising filter, standardized the voxel size with a B-spline interpolation (1 × 1 × 4 mm), applied Z-normalization, and discretized gray levels with a fixed bin size of 16. We extracted 102 radiomics features using the pyradiomics software, which quantitatively characterized the lesion by its shape (n = 14 features), intensity histogram (n = 18 features), and texture (n = 70 features). Specifically, texture features were composed of the gray-level co-occurrence matrix with a distance of 1 voxel (n = 24), gray-level run-length matrix (GLRLM; n = 16), gray-level size-zone matrix (n = 16), and gray-level dependence matrix (GLDM; n = 14) [21].

## 3. Statistical Analysis

Statistical analysis was carried out using Julia and Python (scikit-learn package). We performed a univariate analysis using the Wilcoxon rank-sum test with Bonferroni correction, aiming to identify features with significantly different median values between the ccRCC and oncocytoma groups.

Following a supervised ML approach, we used a stratified cross-validation method with 100 repetitions to obtain statistics of the performance metrics regarding variations in the training dataset. The stratification catered to unbalanced classes. Within each fold, we trained multiple pipelines on the training set and tested their performance on the testing set. Each pipeline consisted of feature normalization, feature selection, and classification. We used either a standard (centered values with unit variance) or robust scaling for normalization. For feature selection, we used a univariate analysis (Wilcoxon rank-sum test), mutual information, or random forest (RF) importance scores. For the classification, we assessed the following algorithms: logistic regression (LR), RF, K-nearest neighbors (KNN), support vector machine (SVM), and AdaBoost. The best performing pipeline was determined using the mean receiver operating characteristic area under the curve (AUC).

Subsequently, we analyzed the importance of features in the classification process within the best performing pipeline, deriving a radiomic signature of the diagnosis. We computed each feature’s selection percentage and the importance scores of features with a selection rate exceeding 75% from the feature selection step. We also calculated Pearson’s correlations among these features to identify groups of correlated features. Finally, we selected a subset of features potentially constituting a radiomic signature of the diagnosis and repeated the classification procedure based on this subset. This constituted a preliminary analysis requiring further validation on an independent dataset.

For the unsupervised approach, we applied a K-means clustering algorithm, initialization kmpp. The optimal cluster number was determined based on the silhouette value.

## 4. Results

A total of 48 MRIs were included, 29 from men, and 19 from women. The mean age was 59.1 (19–83) for patients with RCC and 67.1 (43–88) for patients with oncocytoma. A total of 28 RCC and 19 oncocytoma were included.

The mean age of patients with clear cell renal cell carcinoma was 59.1 years, ranging from 19 to 83, while patients with oncocytoma had a slightly higher mean age of 67.1 years, with a range of 43 to 88. Among the patients with clear cell renal cell carcinoma, 71.4% were male (20 out of 28), whereas 28.6% were female (8 out of 28). In contrast, among patients with oncocytoma, 45% were male (9 out of 20), and 55% were female (11 out of 20). The majority of clear cell renal cell carcinomas were localized in the right kidney, accounting for 64.3% of cases (18 out of 28), while 35.7% were in the left kidney (10 out of 28). For oncocytoma, 55% were in the right kidney (11 out of 20), and 45% were in the left kidney (9 out of 20). When considering the topography of the renal masses, clear cell renal cell carcinomas were often found in the superior pole (50%, 14 out of 28), while oncocytomas were more frequently located in the equatorial pole (65%, 13 out of 20). The average size of clear cell renal cell carcinomas was 28.8 mm, with a range of 15 to 40 mm. In contrast, oncocytomas had an average size of 26.3 mm, also ranging from 15 to 40 mm. Histologically, both clear cell renal cell carcinomas and oncocytomas were often diagnosed through biopsy, with 35.7% and 30% of cases, respectively. Tumorectomy was another common diagnostic method, with 53.5% of clear cell renal cell carcinomas and 30% of oncocytomas. Partial nephrectomy was performed in 10.8% of clear cell renal cell carcinoma cases and in 40% of oncocytoma cases.

The characteristics of the patients are summarized in Table 2.

### 4.1. Univariate Analysis

The univariate analysis incorporating Bonferroni correction (for 102 features, the significance threshold for *p*-value was 0.05/102) did not distinguish any significant features (Figure 2).

### 4.2. Supervised Analysis

The AUC scores from various pipelines are illustrated in Figure 3.

The pipeline that yielded the highest AUC score (0.83) employed several components, including standard normalization, feature selection based on Random Forest (utilizing 100 trees and the Gini impurity criterion for splitting, with a minimum sampling split of 2, a minimum samples leaf of 1, and a maximum number of features equal to the square root of 102), and a Logistic Regression (LR) classifier with L2 regularization (regularization strength parameter C = 1). You can find a comprehensive breakdown of the average performance metrics for this specific pipeline in Table 3, and visual representations are available in Figure 4.

Given that each fold may select different features, we calculated the selection rate for each feature from the top-performing pipeline. The results for a selection rate greater than 75% are depicted in Figure 5.

The top four most frequently selected features were of first order. Following this, we analyzed the importance scores of these features, as computed during the feature selection process across the cross-validations (Figure 6).

This further highlighted the significance of the first-order features in the classification process.

Subsequently, we examined Pearson’s correlations among these features (Figure 7).

We identified two groups of correlated features that were uncorrelated with each other. The first group included first-order features (median, 10th percentile, mean, and root mean); the second group consisted of dependence variance (GLDM), large dependence emphasis (GLDM), and long run emphasis (GLRLM). As a result, we reran the classification procedure based on the best pipeline (excluding the previously retrieved feature selection step) and using only two features: 10th percentile (first-order) and dependence variance (GLDM). This approach yielded a mean AUC of 0.90 (Table 4).

All metrics improved from the previous round. However, since the same data were used to identify and test this signature, its performance should be validated on an independent dataset. Thus, this part of the analysis merely offers an initial step towards formulating a radiomics signature for diagnosis.

### 4.3. Unsupervised Analysis

The K-means algorithm-based unsupervised analysis did not yield significant findings (Figure 8).

When the data were divided into two clusters, a 64% correspondence with the oncocytoma/ccRCC classification was observed, which was less robust than the supervised analysis.

## 5. Discussion

We evaluated the diagnostic potential of radiomics to distinguish between small ccRCC and oncocytoma using the T2 axial sequence on MRI. We employed several pipelines, including standard and robust normalization, univariate analysis, mutual information, RF importance scores for selection, and various ML algorithms such as RF, LR, AdaBoost, KNN, and SVM for classification. The most effective algorithm was determined through a pipeline involving standard normalization, RF-based feature selection, and LR. This method yielded an AUC of 83%, accuracy of 73%, sensitivity of 79%, and specificity of 65%. The training data did not overlap with the validation data, enhancing the robustness of the algorithm.

We used this algorithm to identify the most relevant features for distinguishing between small ccRCC and oncocytoma. We established two groups of uncorrelated features using Pearson’s correlation scores: one encompassing first-order features and the other including texture parameters. After selecting two features from these groups, we developed an algorithm through a pipeline involving standard normalization and LR. This algorithm successfully distinguished between ccRCC and oncocytoma, with an AUC of 90%, accuracy of 77%, sensitivity of 83%, and specificity of 69%.

A recent study demonstrated a radiologic–radiomics ML model capable of distinguishing between benign and malignant renal masses using CT-based radiomics features in various renal tumors and radiological evaluation, achieving sensitivity and specificity rates of up to 91.7% [22]. Other researchers have examined the radiomics of SRMs, a diagnostic challenge in daily radiological practice. Using multiphase contrast enhancement MRI sequences and an RF algorithm on 142 tumors, Hoang et al. reported an accuracy of 79.3%, sensitivity of 67.3%, and specificity of 88.9% in distinguishing small oncocytomas from ccRCCs [16]; they noted that first-order parameters, such as the 10th percentile, could assist in distinguishing ccRCC from oncocytoma.

Coy et al., employing a ML algorithm based on multiphasic CT with 179 lesions, reported an accuracy of 75.4%, sensitivity of 88.3%, and specificity of 43.1% in distinguishing oncocytomas from ccRCCs [17]. Meanwhile, a standard multiparameter MRI-based analysis by Cornelis revealed an accuracy of 77.9% and specificity of 94.2% in distinguishing oncocytomas from ccRCCs. Notable differences in performance in this series were attributable to larger lesion sizes and their use of multiparametric MRI, including double-echo chemical shift, dynamic contrast-enhanced T1- and T2-weighted images, and apparent diffusion coefficient (ADC) maps [23]. They observed that oncocytomas had a higher ADC value but lower signal intensity index in T2-weighted images. The combination of complete central enhancement with contrast inversion on delayed post-contrast T1-weighted images, signal intensity index <2%, and tumor-to-spleen ratio >6% provided 95% specificity for an oncocytoma diagnosis [24].

Canvasser et al. reported an average accuracy of 79%, sensitivity of 78%, and specificity of 80% for the diagnosis of small ccRCC based on an MRI likelihood score, but this was not specific to identifying benign renal masses [25].

Recently, Garnier et al. [26] studied radiomics using CT with a total of 122 patients with 132 renal lesions, including 111 renal cell carcinomas (RCCs) (111/132, 84%) and 21 benign tumors (21/132, 16%). Malignancy was associated to unilaterality (100/111, 90% vs. 13/21, 62%; *p* = 0.02), necrosis (81/111, 73% vs. 8/21, 38%; *p* = 0.02), lower values of tumor/cortex ratio at portal time (0.61 vs. 0.74, *p* = 0.01), and higher variation of tumor/cortex ratio between arterial and portal times (0.22 vs. 0.05, *p* = 0.008). A total of 35 radiomics features were selected, and “intensity mean value” was associated with RCCs in multivariate analysis (OR = 0.99). After ten-fold cross-validation, a C5.0Tree model was retained for its predictive performances, yielding a sensitivity of 95%, specificity of 42%, accuracy of 87%, and AUC of 0.74

In a recent meta-analysis evaluating the use of contrast enhancement ultrasound (CEUS), Tufano et al. [27] reported an accuracy of 0.93 in detecting malignant masses (sensitivity of 0.94, PPV of 0.95, specificity of 0.78, and NPV of 0.73).

Contrast-enhanced ultrasound offers the advantage of real-time visualization of renal tumors through the administration of intravenous contrast agents, facilitating the dynamic evaluation of tumor vascularity and perfusion, as reported in various studies. In the context of renal cell carcinomas (RCCs), prior research has highlighted recurring characteristics. Notably, diffuse heterogeneous enhancement, late washout, and the presence of a perilesional rim-like enhancement, commonly referred to as a pseudocapsule, have been observed in RCCs. Studies have suggested a potential inverse relationship between the presence of pseudocapsules and RCC size, offering valuable diagnostic insights. It is important to note that pseudocapsules are typically absent in cases of hemorrhagic cysts and angiomyolipomas. Additionally, heterogeneous enhancement patterns have been observed across different RCC subtypes, including clear cell, papillary, and chromophobe RCCs, with no significant differences identified among these subtypes. Furthermore, oncocytomas, despite occasionally displaying central stellate scar formations or a spoke–wheel pattern of vascularity, have presented diagnostic challenges in contrast-enhanced ultrasound (CEUS) examinations and have occasionally been misdiagnosed as RCCs. Recently, Tufano et al. [28] studied CEUS on RCC and oncocytoma and reported a significant difference in terms of enhancement (homogeneous vs. heterogeneous), wash-in (fast vs. synchronous/slow), wash-out (fast vs. synchronous/slow), and rim-like enhancement. Heterogeneous enhancement and the presence of a rim-like enhancement had AUCs of 82.5% and 75.3%.

The medical field is now moving towards integrating biology and genomics. Numerous genomic studies have demonstrated tumor heterogeneity in malignant tumors [29], and radiomics is employed to correlate genomic datasets with imaging features [30,31]. For instance, Kocak et al. analyzed 65 ccRCCs to accurately determine the presence or absence of the BABP1 mutation, which indicates a poorer prognosis [20]. Biomarkers on renal tumors may be used for screening, detection, and prognosis [32].

In situations where radiomic features diverge from the pathological findings obtained through biopsy, the necessity for a secondary biopsy arises. This additional biopsy procedure plays a pivotal role in advancing the prospects of personalized interventional medicine and radiology. Radiomic analysis has the potential to identify focal areas of interest within the mass, enhancing the precision of biopsy targeting. Indeed, biopsies occasionally yield negative results, often due to factors like necrosis. Radiomic analysis, however, offers the prospect of refined and more accurate targeting during the biopsy process. By doing so, it allows us to address the challenges posed by negative biopsies, potentially leading to improved diagnostic outcomes and more effective patient care.

Our study had several limitations. First, this was a retrospective study with a relatively small sample size compared to the number of radiomics features extracted, potentially creating selection bias and model overfitting. Second, the segmentation was performed by only one observer and without an inter-observer variability measurement, potentially affecting the results. Third, variability in the MRI parameters might have influenced the robustness and reproducibility of radiomics features [33]. Fourth, we used only the T2 sequence, although some studies have suggested that using more sequences, especially post-gadolinium-weighted sequences, yield better results than the T2 sequence alone [34]. Finally, our model was not tested on an independent cohort; therefore, its generalizability should be evaluated on an independent validation set.

In conclusion, this study demonstrated that ML models incorporating T2-weighted MRI-based radiomics features can help distinguish small ccRCC from oncocytoma, though the results are not superior to standard multiparameter renal MRI. Radiomics does not eliminate the need for percutaneous biopsy. 

## Figures and Tables

**Figure 1 jpm-13-01444-f001:**
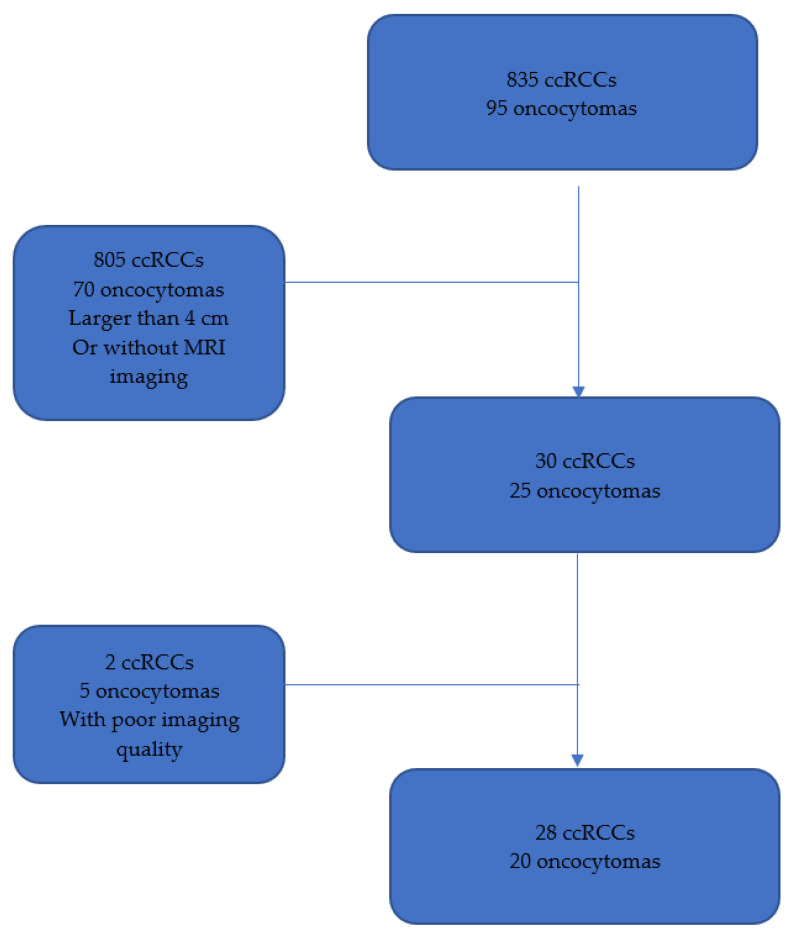
Flowchart of the study inclusion and exclusion criteria. ccRCC, clear cell renal cell carcinoma; MRI, magnetic resonance imaging.

**Figure 2 jpm-13-01444-f002:**
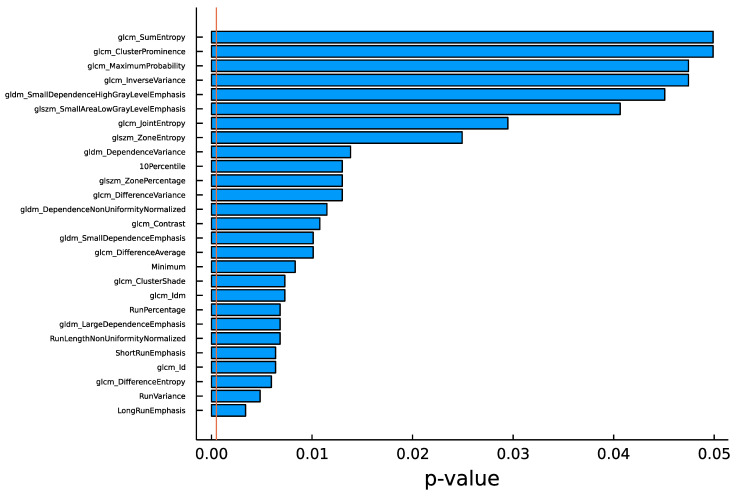
Least significant *p*-values of radiomic features. The vertical line denotes the updated threshold following Bonferroni correction. No features were identified as statistically significant based on univariate analysis.

**Figure 3 jpm-13-01444-f003:**
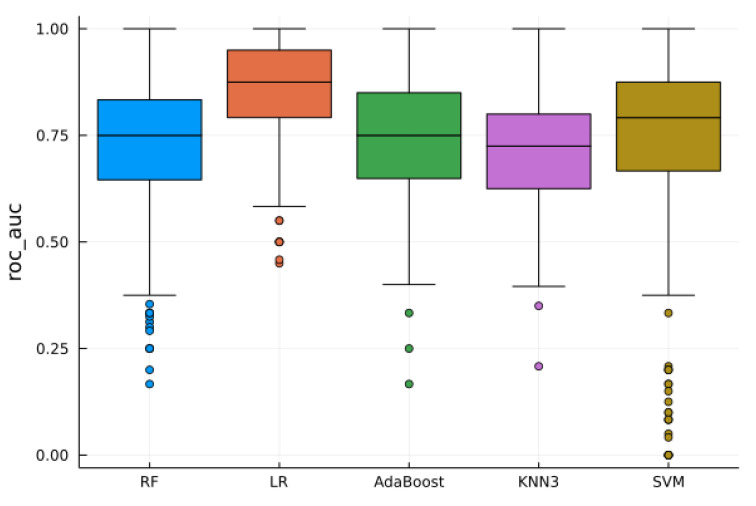
Receiver operating characteristic area under the curve (ROC AUC) scores following standard normalization and univariate analysis feature selection for random forest (RF), logistic regression (LR), AdaBoost, K-nearest neighbor (KNN3), and support vector machine (SVM), based on a stratified five-fold cross-validation with 100 repetitions.

**Figure 4 jpm-13-01444-f004:**
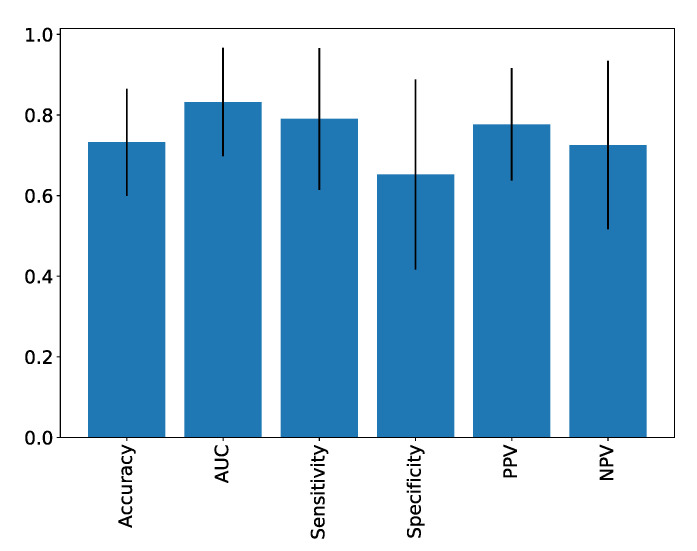
Classification scores of the optimum model (standard normalization, random forest selection, and logistic regression) over 100 repetitions of five-fold cross-validation. AUC, area under the curve; NPV, negative predictive value; PPV, positive predictive value.

**Figure 5 jpm-13-01444-f005:**
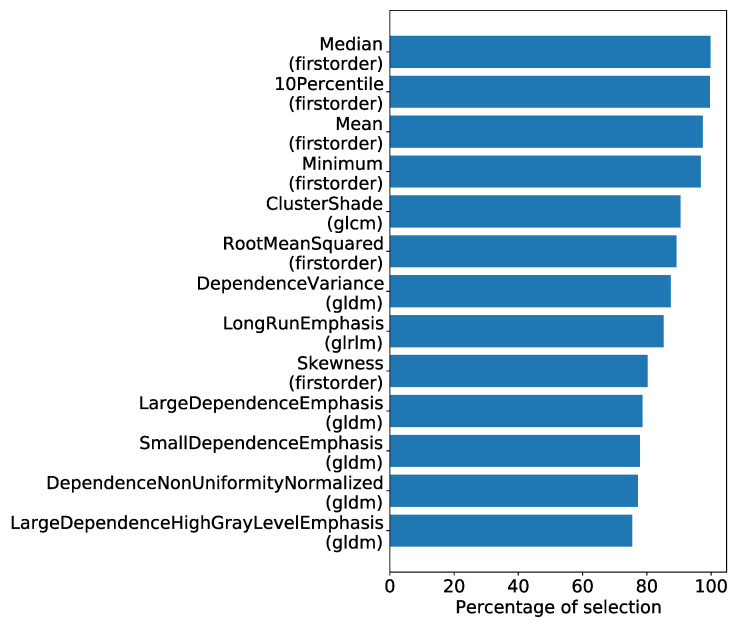
Features with selection rates exceeding 75% using the most effective classification model.

**Figure 6 jpm-13-01444-f006:**
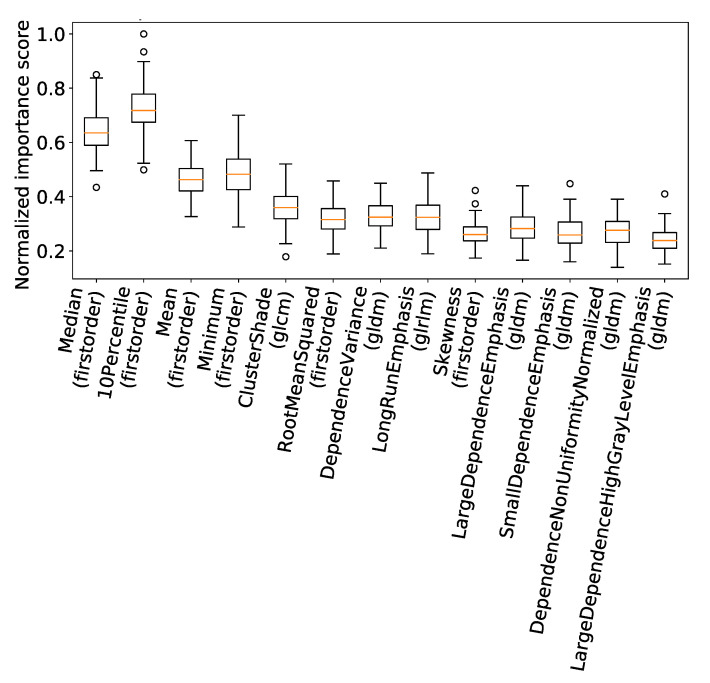
Boxplot of normalized feature importance scores for features with a selection rate exceeding 75%.

**Figure 7 jpm-13-01444-f007:**
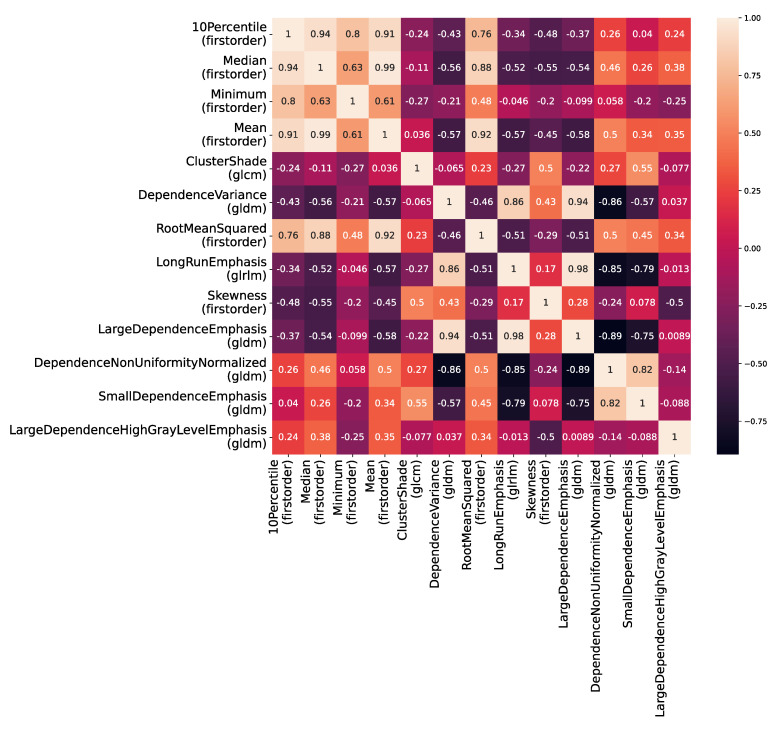
Correlation of features with a selection rate exceeding 75%.

**Figure 8 jpm-13-01444-f008:**
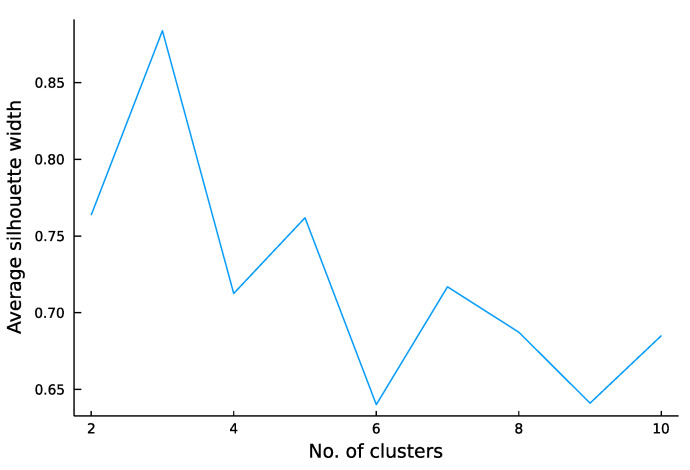
Clustering coefficients for the K-means unsupervised analysis.

**Table 1 jpm-13-01444-t001:** Magnetic resonance imaging system parameters.

	SIEMENS Avanto (1.5T)	Philips Medical Systems Achieva (1.5T)	GE Medical Systems Discovery MR750W (3T)
Sequences	T2	T2	T2
Slices	32	55	30
Thickness (mm)	4	3	4
Intersection gap (mm)	0.4	0.3	0.5
Flip angle (°)	140	90	142
FOV (mm)	380	288	360
Matrix (pixels)	384 × 384	252 × 252	384 × 288
Resolution (mm)	1 × 1	1.14 × 1.14	0.9 × 1.25
TR (ms)	7152.9	2874	12,000
TE (ms)	110	100	160
Duration (s)	170	144	216

TE: echo time; FOV: field of view; TR: repetition time.

**Table 2 jpm-13-01444-t002:** Patient characteristics and tumor localization, topography, average size, and histology.

		Clear Cell Renal Cell Carcinoma (Mean, Range)	Oncocytoma (Mean, Range)
Age (years)		59.1 (19–83)	67.1 (43–88)
Sex (n, %)	Male	20/28 (71.4%)	9/20 (45%)
	Female	8/28 (28.6%)	11/20 (55%)
Localization (n,%)	Right kidney	18/28 (64.3%)	11/20 (55%)
	Left kidney	10/28 (35.7%)	9/20 (45%)
Topography (n,%)	Superior pole	14/28 (50%)	3/20 (15%)
	Equatorial pole	8/28 (29%)	13/20 (65%)
	Inferior pole	6/28 (21%)	4/20 (20%)
Average size (mm)		28.8 (15–40)	26.3 (15–40)
Histology (%)	Biopsy	10/28 (35.7%)	6/20 (30%)
	Tumorectomy	15/28 (53.5%)	6/20 (30%)
	Partial nephrectomy	3/28 (10.8%)	8/20 (40%)

**Table 3 jpm-13-01444-t003:** Performance metrics (mean ± standard deviation) for the best-performing pipeline: standard normalization, random forest-based feature selection, and logistic regression.

	Accuracy	AUC	Sensitivity	Specificity	PPV	NPV
Scores	0.73 ± 0.13	0.83 ± 0.13	0.79 ± 0.18	0.65 ± 0.24	0.78 ± 0.14	0.73 ± 0.21

AUC, area under the curve; NPV, negative predictive value; PPV, positive predictive value.

**Table 4 jpm-13-01444-t004:** Cross-validation performance metrics (mean ± standard deviation) obtained using only two features (10th percentile (first-order) and dependence variance (gray-level dependence matrix)), for the model involving standard normalization and logistic regression.

	Accuracy	AUC	Sensitivity	Specificity	PPV	NPV
Scores	0.77 ± 0.12	0.90 ± 0.10	0.83 ± 0.16	0.69 ± 0.22	0.81 ± 0.12	0.78 ± 0.19

AUC, area under the curve NPV, negative predictive value; PPV, positive predictive value.

## Data Availability

Not applicable.

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
