# Peer review of "Differentiation of Small Clear Renal Cell Carcinoma and Oncocytoma through Magnetic Resonance Imaging-Based Radiomics Analysis: Toward the End of Percutaneous Biopsy"

_jpm, 2023, doi:10.3390/jpm13101444_

Round 1
Reviewer 1 Report
We thank the authors for presenting this manuscript on the Differentiation of Small Clear Renal Cell Carcinoma and Oncocytoma Through Magnetic Resonance Imaging-based Radiomics Analysis.
This is a hot topic in uro-oncology, I would like to congratulate with the authors for the methods and analsyis used.
Main limitation is the retrospective nature and the low sample size.
Below my main suggestions to improve the manuscript
Methods:
- 3 different MRI were used for this study. May this affect the outcome of the study?
- why did authors choose for a supervised and unsupervised analysis, it is not clear, results are diffiuclt to follow. Escpecially this part.
The highest AUC 133 score (0.83) was achieved with a pipeline that incorporated standard normalization, RF- 134 based feature selection (using 100 trees and the Gini impurity criterion for splitting, a min- 135 imum sampling split of 2, a minimal samples leaf of 1, and a maximum number of features 136 of √102 ), and a LR classifier (L2 regularization norm with inverse of regularization 137 strength C = 1). The detailed mean performance metrics for this pipeline are provided in 138 Table 3 (see also Figure 4).
can this be reformulated?
- Please and main chartactheristic of the patients in the first part of the results (not only in table 1.)
Discussion:
- This section is to poor in content, explain more deeply these intresting findings.
A suggestion could be to discuss, in a paragraph, the the results of MRI with the CEUS tool for oncocytoma and for SMR. Below some intresting articles on the topic.
* Qualitative Assessment of Contrast-Enhanced Ultrasound in Differentiating Clear Cell Renal Cell Carcinoma and Oncocytoma
* Diagnostic Performance of Contrast-Enhanced Ultrasound in the Evaluation of Small Renal Masses: A Systematic Review and Meta-Analysis. Diagnostics (Basel). 2022 Sep 25;12(10):2310. doi: 10.3390/diagnostics12102310.
minor English revision is needed
Author Response
We thank the authors for presenting this manuscript on the Differentiation of Small Clear Renal Cell Carcinoma and Oncocytoma Through Magnetic Resonance Imaging-based Radiomics Analysis.
This is a hot topic in uro-oncology, I would like to congratulate with the authors for the methods and analsyis used.
Main limitation is the retrospective nature and the low sample size.
Below my main suggestions to improve the manuscript
Methods:
- 3 different MRI were used for this study. May this affect the outcome of the study?
Authors: Thanks for this comment, we write it already in limitation, Line 368:
« Third, variability in the MRI parameters might have influenced the robustness and reproducibility of radiomics features »
- why did authors choose for a supervised and unsupervised analysis, it is not clear, results are diffiuclt to follow. Escpecially this part.
The highest AUC 133 score (0.83) was achieved with a pipeline that incorporated standard normalization, RF- 134 based feature selection (using 100 trees and the Gini impurity criterion for splitting, a min- 135 imum sampling split of 2, a minimal samples leaf of 1, and a maximum number of features 136 of √102 ), and a LR classifier (L2 regularization norm with inverse of regularization 137 strength C = 1). The detailed mean performance metrics for this pipeline are provided in 138 Table 3 (see also Figure 4).
can this be reformulated?
Authors: We reformulated as :
The pipeline that yielded the highest AUC score (0.83) employed several components, including standard normalization, feature selection based on Random Forest (utilizing 100 trees and the Gini impurity criterion for splitting, with a minimum sampling split of 2, a minimum samples leaf of 1, and a maximum number of features equal to the square root of 102), and a Logistic Regression (LR) classifier with L2 regularization (regularization strength parameter C = 1). Comprehensive breakdown of the average performance metrics for this specific pipeline is reported in Table 3, and visual representations are available in Figure 4
- Please and main chartactheristic of the patients in the first part of the results (not only in table 1.)
Authors: We add this paragraph:
“48 MRI were included, 29 men, and 19 women. Mean age was 59.1 (19–83) for patients with RCC, and 67.1 (43–88) with oncocytoma. 28 RCC and 19 oncocytoma were included.
The mean age of patients with clear cell renal cell carcinoma was 59.1 years, ranging from 19 to 83, while patients with oncocytoma had a slightly higher mean age of 67.1 years, with a range of 43 to 88. Among the patients with clear cell renal cell carcinoma, 71.4% were male (20 out of 28), whereas 28.6% were female (8 out of 28). In contrast, among patients with oncocytoma, 45% were male (9 out of 20), and 55% were female (11 out of 20). The majority of clear cell renal cell carcinomas were localized in the right kidney, accounting for 64.3% of cases (18 out of 28), while 35.7% were in the left kidney (10 out of 28). For oncocytoma, 55% were in the right kidney (11 out of 20), and 45% were in the left kidney (9 out of 20). When considering the topography of the renal masses, clear cell renal cell carcinomas were often found in the superior pole (50%, 14 out of 28), while oncocytomas were more frequently located in the equatorial pole (65%, 13 out of 20). The average size of clear cell renal cell carcinomas was 28.8 mm, with a range of 15 to 40 mm. In contrast, oncocytomas had an average size of 26.3 mm, also ranging from 15 to 40 mm. Histologically, both clear cell renal cell carcinomas and oncocytomas were often diagnosed through biopsy, with 35.7% and 30% of cases, respectively. Tumorectomy was another common diagnostic method, with 53.5% of clear cell renal cell carcinomas and 30% of oncocytomas. Partial nephrectomy was performed in 10.8% of clear cell renal cell carcinoma cases and in 40% of oncocytoma cases.”
Discussion:
- This section is to poor in content, explain more deeply these intresting findings.
A suggestion could be to discuss, in a paragraph, the the results of MRI with the CEUS tool for oncocytoma and for SMR. Below some intresting articles on the topic.
* Qualitative Assessment of Contrast-Enhanced Ultrasound in Differentiating Clear Cell Renal Cell Carcinoma and Oncocytoma
* Diagnostic Performance of Contrast-Enhanced Ultrasound in the Evaluation of Small Renal Masses: A Systematic Review and Meta-Analysis. Diagnostics (Basel). 2022 Sep 25;12(10):2310. doi: 10.3390/diagnostics12102310.
Authors we added this paragraph using this 2 references:
In a recent meta-analysis evaluating using of contrast enhancement ultrasound (CEUS) Tufano et al (27) reported an accuracy of 0.93 in detecting malignant masses (sensitivity of 0.94, PPV of 0.95, specificity of 0.78, and NPV of 0.73).
Contrast-enhanced ultrasound offers the advantage of real-time visualization of renal tumors through the administration of intravenous contrast agents, facilitating the dynamic evaluation of tumor vascularity and perfusion, as reported in various studies. In the context of renal cell carcinomas (RCCs), prior research has highlighted recurring characteristics. Notably, diffuse heterogeneous enhancement, late washout, and the presence of a perilesional rim-like enhancement, commonly referred to as a pseudocapsule, have been observed in RCCs. Studies have suggested a potential inverse relationship between the presence of pseudocapsules and RCC size, offering valuable diagnostic insights. It is important to note that pseudocapsules are typically absent in cases of hemorrhagic cysts and angiomyolipomas. Additionally, heterogeneous enhancement patterns have been observed across different RCC subtypes, including clear cell, papillary, and chromophobe RCCs, with no significant differences identified among these subtypes. Furthermore, oncocytomas, despite occasionally displaying central stellate scar formations or a spoke-wheel pattern of vascularity, have presented diagnostic challenges in contrast-enhanced ultrasound (CEUS) examinations and have occasionally been misdiagnosed as RCCs. Recently Tufano et al (28), studied CEUS on RCC and oncocytoma, and reported a significant difference in terms of enhancement (homogeneous vs. heterogeneous), wash-in (fast vs. synchronous/slow), wash-out (fast vs. synchronous/slow) and rim-like enhancement . Heterogeneous enhancement and the presence of a rim-like enhancement had AUCs of 82.5% and 75.3%.
Reviewer 2 Report
In the current paper draft, authors address the emerging clinical dilemma of preoperative differential diagnosis of small renal masses (SRM), a prerequisite for accurate therapeutic decision-making. Herein, they assess, using ML models, the capability of T2-weighted MRI-based radiomics to distinguish between malignant (ccRCCs) and benign (oncocytomas) SRMs, and implicitly to ultimately better inform real-world surgical indications. Thus, the chosen research topic is of great interest currently, within the overarching aim of identifying implementable non-invasive clinical tools for obtaining a reliable preoperative risk assessment in renal tumors, known for their unpredictable clinical behavior. However, the current paper also has some limitations:
1. Introduction: This chapter seems underdeveloped and should provide a more in-depth analysis of currently available data on other non-invasive diagnostic tools for RCCs, i.e. biomarkers, focusing on the advantages of imaging over biological assays and highlighting the unreliability, variability and overall pitfalls of the plethora of exciting, yet superficially documented, emerging proposals of novel key molecular targets in RCCs. Moreover, additional attention should provided to the use of AI in assessing the more readily available ultrasound data, for the differential diagnosis of urological malignancies (https://doi.org/10.3390/curroncol29060336).
2. The study cohort is quite limited and spans a large timeframe clinically. This represents a significant limitation. The implications of these limitations must be more clearly analyzed.
3. Overall, the paper suffers from the same contextual limitation as many early papers in an emerging, novel field, lack of validation and meek overall impact of results.
acceptable.
Author Response
In the current paper draft, authors address the emerging clinical dilemma of preoperative differential diagnosis of small renal masses (SRM), a prerequisite for accurate therapeutic decision-making. Herein, they assess, using ML models, the capability of T2-weighted MRI-based radiomics to distinguish between malignant (ccRCCs) and benign (oncocytomas) SRMs, and implicitly to ultimately better inform real-world surgical indications. Thus, the chosen research topic is of great interest currently, within the overarching aim of identifying implementable non-invasive clinical tools for obtaining a reliable preoperative risk assessment in renal tumors, known for their unpredictable clinical behavior. However, the current paper also has some limitations:
- Introduction: This chapter seems underdeveloped and should provide a more in-depth analysis of currently available data on other non-invasive diagnostic tools for RCCs, i.e. biomarkers, focusing on the advantages of imaging over biological assays and highlighting the unreliability, variability and overall pitfalls of the plethora of exciting, yet superficially documented, emerging proposals of novel key molecular targets in RCCs. Moreover, additional attention should provided to the use of AI in assessing the more readily available ultrasound data, for the differential diagnosis of urological malignancies (https://doi.org/10.3390/curroncol29060336).
Thanks, we added on discussion: Biomarkers on renal tumors may be used for screening, detection and prognosis (32).
- The study cohort is quite limited and spans a large timeframe clinically. This represents a significant limitation. The implications of these limitations must be more clearly analyzed.
Authors: thanks for this comment. We have analyzed limitation on last paragraph of the discussion, and we wrote in conclusion’ Radiomics does not eliminate the need for percutaneous biopsy’
- Overall, the paper suffers from the same contextual limitation as many early papers in an emerging, novel field, lack of validation and meek overall impact of results.
Authors: of course.
Round 2
Reviewer 1 Report
I have checked the revisioned article. The authors cleared my main concerns. The article may be accepted now.Reviewer 2 Report
Revised according to comments.